# Predicting the Order of Upcoming Tokens Improves Language Modeling

## Abstract

Multi-Token Prediction (MTP) has been proposed as an auxiliary objective to improve next-token prediction (NTP) in language model training but shows inconsistent improvements, underperforming in standard NLP benchmarks. We found MTP's exact future token prediction to be too difficult as an auxiliary loss. Instead, we propose Token Order Prediction (TOP), which trains models to order upcoming tokens by their proximity using a learning-to-rank loss. TOP requires only a single additional unembedding layer compared to MTP's multiple transformer layers. We pretrain models of 340M, 1.8B, and 7B parameters using NTP, MTP, DeepSeek MTP and TOP objectives. The results of eight standard NLP benchmarks show that TOP overall outperforms NTP, MTP, and DeepSeek MTP even at scale. On the synthetic star graph task, TOP enables pathfinding on graphs where NTP and MTP fail.

## 1 Introduction

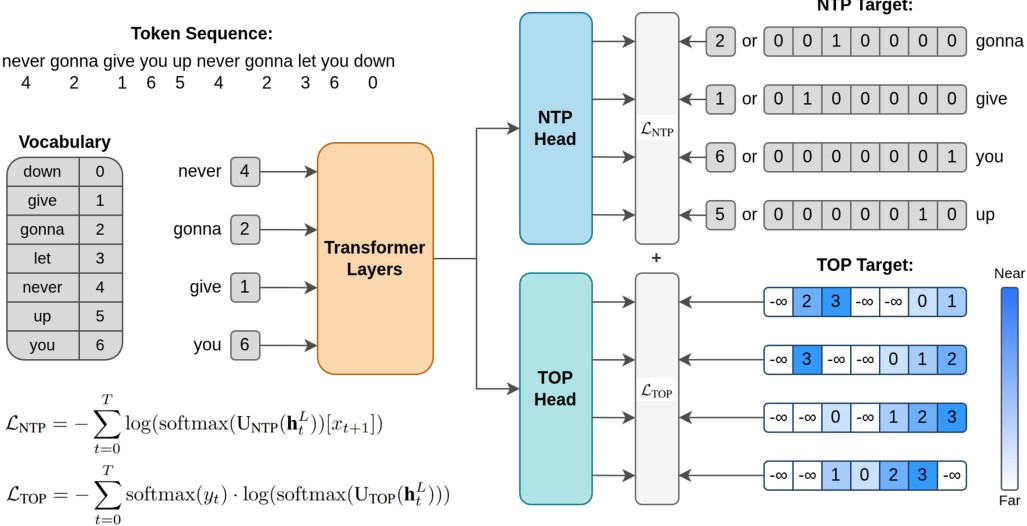

Figure 1: An overview of Token Order Prediction (TOP). Given an input token sequence, a vocabulary, a sequence length of 4 and window size of 4, a TOP target sequence is constructed via Algorithm 1. The output hidden representation of the final layer goes to two separate unembedding heads for NTP and TOP. The final loss to optimize is a sum of the NTP and TOP loss.

Current large language models (LLMs) are trained to predict the next token in a sequence during training, an unsupervised learning task often referred to as next-token prediction (NTP). Although simple, NTP has been very successful in creating powerful language models that can solve complex tasks and even reason over their context.

However, NTP has received various criticisms in recent years. A notable argument by LeCun (2024) claims that NTP at inference time accumulates errors over every time step and inevitably falls off greatly in accuracy. This was however refuted by Bachmann & Nagarajan (2024), in which they argue that the main issue of NTP lies not in inference time error accumulation; rather, that teacher-forcing is unable to learn an accurate next-token predictor in the first place.

Building off ideas such as ProphetNet (Qi et al., 2020), Multi-Token Prediction (MTP) (Gloeckle et al., 2024) has emerged as a relatively successful auxiliary learning task to improve NTP in LLM training. MTP adds multiple heads to the end of a transformer that each predict a different offset of tokens ahead. All MTP heads share the same trunk of transformer layers, with the hope that having these auxiliary heads leads to the model learning better internal representations that are considerate of not only the next immediate token, but also future tokens that may come after it. It has been shown that MTP improves performance of language models on certain generative tasks that require some level of look-ahead, such as coding and summarization. This method was used in the training of DeepSeek-V3 (DeepSeek-AI et al., 2024), although with sequential instead of parallel MTP heads.

However, the original MTP paper (Gloeckle et al., 2024) showed that MTP does not generally improve language modeling performance on every standard NLP task, as evident from the downstream task performance seen in their Appendix G. Important to note is that MTP does not seem to work on smaller models. In generative tasks, MTP harms performance and only starts to gain advantage over NTP for models with over 1-3 billion parameters. Additionally, the number of future tokens to predict is a hyperparameter that needs to be set before training. This is critical because increasing the future token count requires adding more heads, which adds more parameters to train and more compute required. Furthermore, the paper shows that increasing the number of future tokens does not guarantee better performance even on the benefiting tasks, where it is shown that 4 future tokens performs better than 8 in coding.

We aim to improve upon MTP by introducing a different auxiliary training objective with the same goal as MTP: enhancing next-token prediction performance through better internal representations. However, instead of exactly predicting multiple future tokens, we propose that a better training objective is to predict the order of upcoming tokens in the sequence with a learning-to-rank loss. In this paper, we contribute the following:

1. We introduce Token Order Prediction (TOP), a novel auxiliary training loss in addition to NTP to improve language modeling in general.

2. For each of the four training strategies NTP, MTP, DeepSeek MTP (DS-MTP) and TOP, we pretrain language models with sizes of 340M, 1.8B, and 7B parameters on up to 104B tokens.

3. We evaluate these models on standard NLP benchmarks and show that TOP improves upon NTP, MTP, and DS-MTP even at scale. Additionally, the synthetic star graph pathfinding task shows TOP can solve lookahead problems that NTP, MTP, and DS-MTP cannot.

## 2 BACKGROUND

Next-token prediction (NTP) is the standard training objective for present-day language models. This task is learned by optimizing the cross-entropy loss over the sequence length. Given sequence length $T$, model dimension $D$, vocabulary size $V$ and $\mathbf{x} = \{x_0, \ldots, x_{T+1} \mid x_i \in \mathbb{Z}\}$ as the input token sequence, this loss is written as

$$\mathcal{L}_{\text{NTP}} = -\sum_{t=0}^{T} \log(\mathrm{P}_\theta(x_{t+1}|x_{0:t})) \tag{1}$$

where $\mathrm{P}_\theta$ is the output probability given by the language model with parameters $\theta$. The probability of the next token $x_{t+1}$ given this model is written as

$$\mathrm{P}_\theta(x_{t+1}|x_{0:t}) = \text{softmax}(\mathrm{U}_{\text{NTP}}(\mathbf{h}_t^L))[x_{t+1}] \tag{2}$$

where the hidden representation $\mathbf{h}_t^L \in \mathbb{R}^D$ is generated by a transformer up to the final layer $L$ conditioned on $x_{0:t}$, and the NTP head $\mathrm{U}_{\text{NTP}} : \mathbb{R}^D \to \mathbb{R}^V$ is a linear unembedding layer to project $\mathbf{h}_t^L$ onto the vocabulary. The probability is taken at the index of the target token $[x_{t+1}]$.

Multi-Token Prediction (MTP) (Gloeckle et al., 2024) was proposed as an architectural modification that adds additional MTP heads[1] in the form of parallel, singular transformer layers that each output a future token prediction at offset positions. Given $N$ as the number of future tokens to predict (including the next token), the MTP loss can be written as

$$\mathcal{L}_{\text{MTP}} = -\sum_{t=0}^{T} \log(\mathrm{P}_\theta(x_{t+1:t+N}|x_{0:t})) = -\sum_{t=0}^{T}\sum_{n=1}^{N} \log(\mathrm{P}_\theta(x_{t+n}|x_{0:t})) \tag{3}$$

---

[1]Disambiguation: Architecturally, MTP heads are transformer blocks, as we follow the terminology from Gloeckle et al. (2024). Meanwhile, NTP head and TOP head are linear unembedding layers.

If we define $\mathbf{h}_t^{L-1}$ as the hidden representation before the last transformer layer and have $\mathrm{F}_i$ for $i = 1, .., N$ as the MTP heads in the form of singular transformer layers for each future token, and all heads share the same unembedding layer or NTP head $\mathrm{U}_{\mathrm{NTP}}$, then

$$\mathrm{P}_\theta(x_{t+n}|x_{0:t}) = \mathrm{softmax}(\mathrm{U}_{\mathrm{NTP}}(\mathrm{F}_n(\mathbf{h}_t^{L-1})))[x_{t+n}] \tag{4}$$

MTP promises better performance on generative tasks such as coding and summarization that benefit from the look-ahead nature of MTP. MTP also allows the model to do a form of self-speculative decoding, which speeds up inference to some degree. However, as mentioned earlier, MTP does not seem to improve overall language modeling performance on downstream tasks other than those mentioned above, struggling on standard NLP benchmarks.

There have also been other MTP variants such as the one used for training DeepSeek V3 (DeepSeek-AI et al., 2024). We refer to this variant as DS-MTP. Unlike the original MTP, DS-MTP arranges the MTP heads sequentially. Each head after the first one, receives concatenated, normalized embeddings from the original tokens up to that offset. We define $\mathrm{E} : \mathbb{R}^V \to \mathbb{R}^D$ as the embedding layer also shared with the main transformer trunk.

$$h_t^{L-1+n} = \begin{cases} \mathrm{F}_n(\mathbf{h}_t^{L-1}) & \text{if } n = 1 \\ \mathrm{F}_n([\mathrm{RMSNorm}(\mathbf{h}_t^{L-1+n-1}); \mathrm{RMSNorm}(\mathrm{E}(x_{0:t+n-1}))] & \text{if } 2 \leq n \leq N \end{cases} \tag{5}$$

$$\mathrm{P}_\theta(x_{t+n}|x_{0:t+n-1}) = \mathrm{softmax}(\mathrm{U}_{\mathrm{NTP}}(\mathbf{h}_t^{L-1+n})[x_{t+n}] \tag{6}$$

$$\mathcal{L}_{\mathrm{DS\text{-}MTP}} = \sum_{t=0}^{T} \sum_{n=1}^{N} \log(\mathrm{P}_\theta(x_{t+n}|x_{0:t})) \tag{7}$$

Notably, DeepSeek V3 only used $N = 2$ for their MTP training, which means the model only learns to predict the next two tokens. This might be related to a key point of our paper, which is that MTP is too difficult an objective to be an effective auxiliary loss to NTP, especially for larger look-ahead values $N$.

## 3 MOTIVATION

Our hypothesis for why MTP only partially improves language modeling is that MTP is too difficult as a learning objective. If we look at the original MTP paper (Gloeckle et al., 2024), there are two empirical results that support this argument. First, MTP does not improve performance of small language models on generative tasks, such as coding. This suggests that a certain capability threshold is required for MTP's multi-token modeling to be effective, which they observe to be in the 1B-3B parameter range. Second, increasing the number of future tokens in MTP does not guarantee better performance overall. The ideal number of future tokens varies across different tasks. Not only does this make it difficult to determine the optimal number beforehand, it also indicates that there are thresholds of look-ahead distance where the difficulty of prediction starts to hurt learning instead of helping it.

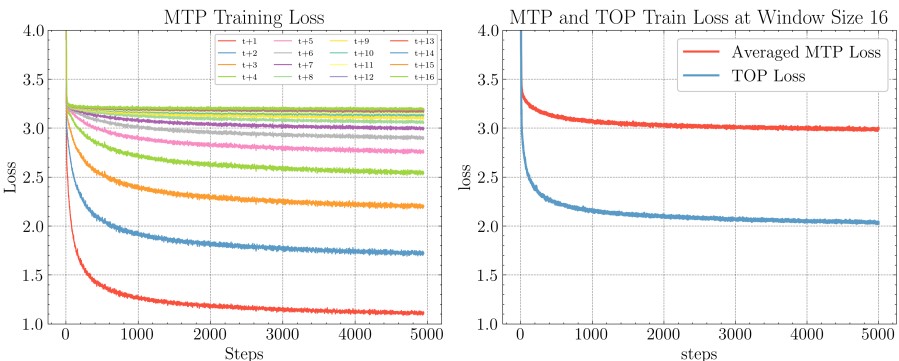

Figure 2: Left: Training loss of a MTP transformer with 16 MTP heads predicting tokens at $t + 1, ..., t + 16$ offsets. Right: Training loss of the MTP model averaged over all 16 heads, compared to the training loss of the TOP model with window size 16.

To illustrate our argument, we train a small 16M parameter transformer with 16 MTP heads and visualize the training loss of each MTP head in Figure 2. A clear pattern emerges where the losses of predicting the tokens at positions $t + 1, ..., t + 16$ arrange themselves from bottom to top. Each future token farther away significantly worsens in loss compared to the immediate next token loss and shows a decreased rate of loss descent, indicating the difficulty of exactly predicting far ahead. We believe that relaxing this MTP objective will make it more useful as an auxiliary loss. Compared to a similarly sized model with the TOP objective at window size 16, we see that the TOP loss is lower.

## 4 METHOD

We propose Token Order Prediction (TOP), a novel auxiliary training loss for language modeling. Given a sequence of tokens $\mathbf{x} = \{x_0, \ldots, x_T \mid x_i \in \mathbb{Z}\}$, we construct a TOP target sequence $\mathbf{y} = \{y_0, \ldots, y_T \mid y_i \in \mathbb{Z}^V\}$ where $V$ is a vocabulary size, in which every index contains a score ranking each token in the vocabulary based on their order in the sequence $\mathbf{x}$, going in descending order from closest to furthest first appearance after $x_t$. We also introduce a hyperparameter, window size, within which the token order is evaluated. To better understand this target sequence, please refer to the pseudocode in Algorithm 1 and the visualization in Figure 1 on how to construct it. In practice, we have an optimized Triton kernel for this function that creates the target sequence on the fly during training and practically incurs no overhead. Alternatively, one could also pre-process an entire dataset beforehand.

---

**Algorithm 1** Convert a token sequence to a TOP target sequence

---

    **Goal:** For each position, compute a "proximity" score to the next occurrence of every token within a window of size $W$.

**Require:** Token sequence $\mathbf{x}$ of length $T + W$, vocab size $V$, window size $W$.

**Ensure:** Tensor $\mathbf{y}$ of shape $(T, V)$

 1: Initialize $\mathbf{y} \leftarrow -\infty$                             ▷ By default, tokens not appearing in the sequence are set to $-\infty$

 2: Initialize $\mathbf{n}[v] \leftarrow T + W$ for all $v \in [0, V - 1]$               ▷ $\mathbf{n}$ keeps track of next occurrence of every token

 3: **for** $t \leftarrow T + W - 1$ down to $0$ **do**                       ▷ Iterate backwards through sequence

 4:     **if** $\mathbf{x}[t]$ is valid **then**                             ▷ Check if token $\mathbf{x}[t]$ is in vocabulary

 5:         $\mathbf{n}[\mathbf{x}[t]] \leftarrow t$                           ▷ Record most recent position of token $\mathbf{x}[t]$

 6:     **end if**

 7:     **if** $t < T$ **then**                                    ▷ Length of output is only $T$

 8:         **for** each $v \in [0, V - 1]$ **do**                 ▷ Update each token in the vocabulary

 9:             $d \leftarrow \mathbf{n}[v] - t$              ▷ $d$ equals steps ahead until next occurrence of token $v$

 10:            **if** $0 < d \leq W$ **then**          ▷ If the distance to next occurrence of token $v$ is within the window

 11:                $\mathbf{y}[t, v] \leftarrow W - d$        ▷ Closer tokens get assigned a larger proximity score, with maximum $W$

 12:            **end if**

 13:         **end for**

 14:     **end if**

 15: **end for**

---

To train the model to order upcoming tokens as in the target sequence, we borrow a loss formulation from the learning-to-rank literature (Pobrotyn et al., 2020), more specifically from ListNet (Cao et al., 2007). This listwise ranking loss is formulated as the distance between the top-one probability of two lists of scores, where the distance metric is cross-entropy. The TOP auxiliary loss is defined as follows:

$$\mathcal{L}_{\text{TOP}} = -\sum_{t=0}^{T} \text{softmax}(y_t) \cdot \log(\text{softmax}(\text{U}_{\text{TOP}}(\mathbf{h}_t^L))) \tag{8}$$

Note that $\text{softmax}(\text{U}_{\text{TOP}}(\mathbf{h}_t^L))$ is not a probability distribution by definition, hence why we do not write it as $\text{P}_\theta$. The correct way to think about it is to view $\text{U}_{\text{TOP}}(\mathbf{h}_t^L)$ as the model prediction of the ranking in the form of proximity scores, and $\text{softmax}(\mathbf{y}) \cdot \log(\text{softmax}(\hat{\mathbf{y}}))$ as the ranking loss defined in ListNet. Also note that there are no additional transformer layers needed for TOP. There is however an additional linear unembedding layer $\text{U}_{\text{TOP}} : \mathbb{R}^D \to \mathbb{R}^V$ in parallel to the NTP head. Both unembedding heads $\text{U}_{\text{NTP}}$ and $\text{U}_{\text{TOP}}$ receive the same hidden state $\mathbf{h}_t^L$, which is the output of the final transformer layer. We refer to these unembedding layers as NTP head and TOP head, respectively. The final loss being optimized is simply a sum of the NTP loss and the TOP loss:

$$\mathcal{L} = \mathcal{L}_{\text{NTP}} + \mathcal{L}_{\text{TOP}} \tag{9}$$

Through this specific target sequence formulation and ranking loss function, the model is expected to learn an internal representation that can approximately *construct* the future sequence by returning the most probable order of upcoming tokens. This is expected to be an easier task than trying to exactly predict a future token at some offset. It is also much more scalable compared to MTP, requiring only one additional unembedding matrix. While this matrix can be large, no additional parameters are needed even when adjusting window size, unlike MTP where each additional future token requires a full extra transformer layer. Specifically, while a TOP head requires $DV$ extra parameters, $N$ number of MTP heads require $N(4D^2 + 12D^2 + 2D)$ assuming a standard transformer block with MLP hidden size $4D$ and 2 RMSNorms. Moreover, the cost of a single unembedding layer gets amortized as the model size scales up. In practice,

Table 1: Training configuration and hyperparameters for {340M, 1.8B, 7B} models.

| Parameter | Value | Parameter | Value |
|---|---|---|---|
| Hidden size | {1024, 2048, 4096} | Optimizer | AdamW |
| Num. layers | {24, 32, 30} | Learning rate | {3e-4, 2e-4, 1.2e-4} |
| Num. heads | {16, 32, 32} | LR schedule | Cosine |
| Num. KV heads | {16, 32, 8} | Minimum LR (%) | 10% |
| Sequence length | 4096 | Warmup steps | {1000, 2000, 2000} |
| RoPE $\theta$ | 10,000 | Device batch size | {16, 16, 8} |
| Tied embeddings | False | Grad. accum. steps | {1, 1, 2} |
| Vocab size | 32,000 | Global batch size | 128 |
| TOP window size | 4096 | Training steps | {100k, 200k, 200k} |
| MTP/DS-MTP num. future tokens | 4 | Gradient clip max. | 1.0 |

we use a fused Triton kernel that readily performs both the unembedding and loss calculation block-wise in one pass, thus making the overhead minimal. This kernel is a modification to fused linear cross-entropy loss kernels from Yang & Zhang (2024), resulting in the same performance as the non-modified version.

We find that we do not need additional transformer blocks like MTP for TOP because both the NTP and TOP heads are mainly aligned on the same objective: assigning the highest score to the next token. Although it is possible to train a language model on only the TOP objective, the resulting model will only be able to do greedy generation. An NTP head is still needed for non-greedy, probability sampling-based inference. At inference time, we remove the TOP head and use only the NTP head, making the model equivalent to the original transformer architecture.

## 5 EXPERIMENTS AND RESULTS

### 5.1 GENERAL LANGUAGE MODELING

We train transformer models for the training methods NTP, MTP, DS-MTP, and TOP in 3 sizes each: 340M, 1.8B, and 7B. These sizes are an approximate naming scheme; each model of each training method will have slightly different parameter counts. We try to match the parameter count at training time, excluding embedding parameters. This means that by setting the MTP or DS-MTP number of future tokens to 4, the shared trunk will be reduced by 3 layers to account for the added MTP heads, as is done in the original MTP paper. We train all models on the sample-100BT subset of FineWeb-Edu (Lozhkov et al., 2024). The 340M models are trained on 52B tokens, while the 1.8B and 7B models are trained on 104B tokens. We use the Flame framework and flash-linear-attention (Yang & Zhang, 2024) repository to implement and train our models. The full training configuration and hyperparameters for all model sizes are detailed in Table 1. We want the TOP window size to be as large as possible but still tractable to compute the target sequence with. Here, we set it to be equal to the sequence length, which means Algorithm 1 will receive an input with twice the sequence length.

We evaluate our models on eight standard NLP benchmarks: ARC challenge (Clark et al., 2018), Lambada (Paperno et al., 2016), PIQA (Bisk et al., 2020), SciQ (Welbl et al., 2017), Social IQa (Sap et al., 2019), TriviaQA (Joshi et al., 2017), NaturalQuestions Open (Kwiatkowski et al., 2019), and HellaSwag (Zellers et al., 2019), with full results presented in Table 2. Across all model sizes, TOP shows overall better performance over MTP, DS-MTP, and the baseline NTP models on most tasks.

Our reproduction of MTP shows smaller MTP models achieve competitive results. This finding complements the original MTP paper, which did not report on models smaller than 7B on the standard NLP benchmarks. Consistent with the original study however, the 7B MTP model underperforms in these tasks. While the MTP paper suggests that it scales effectively on coding tasks, our findings indicate that this scalability does not extend to non-coding tasks. In contrast, our TOP model improves in performance as it scales to 7B and surpasses the 7B NTP and MTP baseline. This suggests that in more general tasks, TOP performs and scales better than MTP. We also do not see an improvement in performance from DS-MTP compared to MTP in our reproduction.

We also report the training loss recorded only on the NTP head of each model (i.e. the first MTP head for MTP models). Interestingly, TOP exhibits a higher training loss than NTP, yet achieves lower Lambada perplexity and better benchmark scores. We argue that TOP may act as a regularizer, mitigating overfitting on the limited FineWeb-Edu subset we used for training. However, this claim may need further investigation.

Table 2: General language modeling evaluation results of NTP vs MTP vs DS-MTP vs TOP on standard NLP benchmarks. We report the NTP head only final training loss, the accuracy and perplexity on Lambada, the normalized accuracy on HellaSwag, ARC Challenge, PIQA, and SciQ, the accuracy on Social IQa, and the exact match score on NaturalQuestions Open and TriviaQA. Green/red values indicate the performance difference from the NTP baseline.

| Size | Model | NTP Loss | Lambada Acc. ↑ | Lambada PPL ↓ | HellaSwag N. Acc. ↑ | ARC N. Acc. ↑ | PIQA N. Acc. ↑ | SciQ N. Acc. ↑ | Social IQa Acc. ↑ | NQ Open E.M. ↑ | TriviaQA E.M. ↑ |
|---|---|---|---|---|---|---|---|---|---|---|---|
| 340M | NTP | 2.39 | 36.35 | 30.34 | 42.53 | 28.84 | 66.65 | 74.90 | 39.82 | 1.94 | 4.93 |
| | MTP | 2.47 | 35.32 -1.03 | 35.31 +4.96 | 42.73 +0.20 | 29.86 +1.02 | 66.49 -0.16 | 77.40 +2.50 | 39.00 -0.82 | 2.35 +0.42 | 2.55 -2.37 |
| | DS-MTP | 2.50 | 34.66 -1.69 | 40.99 +10.65 | 40.29 -2.24 | 27.56 -1.28 | 63.76 -2.88 | 73.70 -1.20 | 37.92 -1.89 | 0.36 -1.58 | 0.87 -4.05 |
| | TOP | 2.40 | 37.07 +0.72 | 28.76 -1.58 | 43.57 +1.04 | 29.35 +0.51 | 67.57 +0.92 | 79.80 +4.90 | 39.00 -0.82 | 2.22 +0.28 | 4.37 -0.55 |
| 1.8B | NTP | 2.06 | 49.58 | 11.38 | 60.05 | 38.65 | 73.50 | 86.40 | 41.56 | 4.54 | 11.85 |
| | MTP | 2.14 | 47.93 -1.65 | 13.69 +2.31 | 58.29 -1.76 | 40.61 +1.96 | 73.07 -0.44 | 87.20 +0.80 | 42.12 +0.56 | 4.46 -0.08 | 15.98 +4.13 |
| | DS-MTP | 2.14 | 48.71 -0.87 | 13.32 +1.94 | 57.48 -2.57 | 40.44 +1.79 | 71.87 -1.63 | 86.40 | 42.84 +1.28 | 4.21 -0.33 | 12.06 +0.21 |
| | TOP | 2.08 | 50.34 +0.76 | 11.19 -0.19 | 60.45 +0.40 | 42.32 +3.67 | 74.16 +0.65 | 87.90 +1.50 | 42.53 +0.97 | 5.37 +0.83 | 18.93 +7.07 |
| 7B | NTP | 1.88 | 55.89 | 7.97 | 67.44 | 45.65 | 76.99 | 88.60 | 44.37 | 7.31 | 24.28 |
| | MTP | 1.95 | 53.13 -2.76 | 8.99 +1.03 | 65.85 -1.58 | 45.56 -0.09 | 75.73 -1.25 | 89.30 +0.70 | 44.11 -0.26 | 7.40 +0.08 | 23.36 -0.92 |
| | DS-MTP | 1.96 | 55.62 -0.27 | 8.52 +0.55 | 66.03 -1.40 | 44.37 -1.28 | 75.79 -1.20 | 88.70 +0.10 | 43.76 -0.61 | 6.57 -0.75 | 18.54 -5.73 |
| | TOP | 1.89 | 57.03 +1.14 | 7.64 -0.32 | 68.73 +1.29 | 46.42 +0.77 | 76.39 -0.60 | 91.60 +3.00 | 43.91 -0.46 | 7.70 +0.39 | 30.90 +6.63 |

## 5.2 THE STAR GRAPH TASK

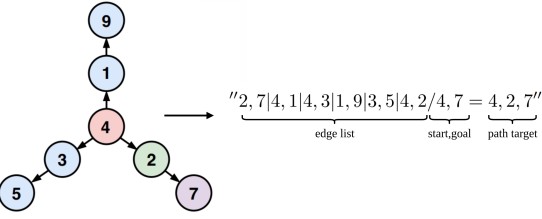

Figure 3: Illustration of a star graph training sample with $d = 3$ and $l = 3$ due to Bachmann & Nagarajan (2024).

In addition to general language modeling, we also evaluate TOP on a synthetic task, the star graph pathfinding problem. It was proposed by Bachmann & Nagarajan (2024) to highlight the weakness of training with NTP where the model fails to learn the correct look-ahead solution. The task is to find a path from a starting node to a goal node, given a star graph $G(d, l)$ with $d$ paths branching from the starting node and path length $l$. Refer to Figure 3 for an example training sample of this task. We train standard transformers with 8 layers, embedding size 384, and 6 attention heads for 6 training objectives: NTP, TOP, MTP-2, MTP-4, DS-MTP-2, and DS-MTP-4. The MTP and DS-MTP models have different number of future tokens i.e. two and four MTP heads. Note that in the case of MTP and DS-MTP in this task, we only subtract one layer from the main trunk and add the MTP heads on top because the models are too small, resulting in MTP models with larger parameter counts compared to NTP and TOP. We train these models on 4 star graph setups: $G(3, 3)$, $G(3, 5)$, $G(5, 3)$, and $G(5, 5)$ with $N = 30$ which means each node label is sampled uniformly from 30 labels i.e. tokens. In each setup we generate 300,000 training samples and 10,000 test samples and train for 100 epochs to convergence. We use a batch size of 4096, learning rate of 0.003 with warmup of 1500 and cosine decay down to a learning rate of 0.001 for all models.

We present the results of the star graph task in Table 3 and Figure 4. Similar to the results in (Bachmann & Nagarajan, 2024), NTP performs poorly across the board and fails to learn the appropriate look-ahead mechanism. We also observe that the effectiveness of MTP is dependent on its number of heads. For instance, a configuration with four heads is effective, while one with only two is insufficient, particularly for graphs with longer paths. However, the TOP model demonstrates the most robust performance. On the $G(5, 5)$ star graph, the TOP model is the only model with perfect test set accuracy, where even the MTP and DS-MTP model with 4 heads fail.

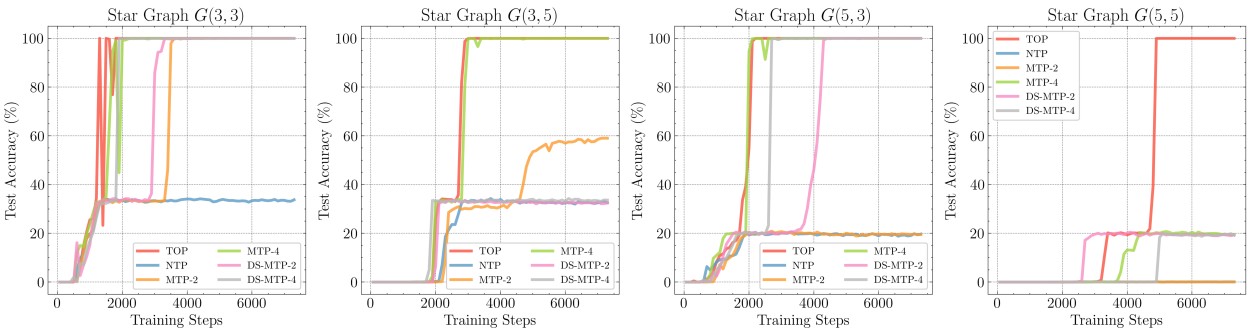

Figure 4: Test set accuracy during training of NTP, TOP and MTP models on the star graph pathfinding task.

Table 3: Parameter count and test set accuracy of star graph task models.

| Model | Param. Count | $G(3,3)$ | $G(3,5)$ | $G(5,3)$ | $G(5,5)$ |
|---|---|---|---|---|---|
| NTP | 14.22M | 33.77 | 32.53 | 19.49 | 0.06 |
| MTP-2 | 16.00M | **100** | 59.02 | 19.58 | 0.05 |
| MTP-4 | 19.55M | **100** | **100** | **100** | 19.53 |
| DS-MTP-2 | 16.59M | **100** | 32.46 | **100** | 19.20 |
| DS-MTP-4 | 20.73M | **100** | 33.62 | **100** | 19.26 |
| TOP | 14.24M | **100** | **100** | **100** | **100** |

## 5.3 SELF-SPECULATIVE DECODING

Speculative decoding Stern et al. (2018) is a technique for accelerating inference by first using a smaller model to generate predictions and then employing the original model as a validator. In MTP, all of the heads can be used simultaneously to predict future tokens, with each head generating one token. The predicted tokens are then validated in a second forward pass using the original model. Because MTP uses the same model, the technique is referred to as self-speculative decoding. This requires performing inference twice: once for generation and once for validation.

Although TOP is intended to be used only at training time to improve learning, we also explore the possibility of using the TOP head for self-speculative decoding. To do this, we construct a future sequence by ordering the TOP head's predicted tokens by proximity score, appending them to the input, and running another forward pass to check the longest common prefix i.e. acceptance rate with the NTP head's predictions. Importantly, the acceptance rate or average number of accepted tokens does not imply a directly equivalent inference speed-up because two forward passes are required.

We take all TOP, MTP, and DS-MTP models and evaluate their self-speculative decoding potential on texts of different domains. For each domain, we take 5000 random snippets of text and calculate the average number of accepted tokens per validation forward pass. We present the results in Table 4. Evidently, TOP does not perform as well as MTP nor DS-MTP for self-speculative decoding. A maximum acceptance rate of 1.54 on the 7B TOP model might translate to a 1.2-1.4x inference speed-up, depending on the implementation. Meanwhile the 7B MTP model goes up to 2.49 acceptance rate, still slightly below the numbers reported in the original paper given 4 MTP heads, while the DS-MTP 7B model achieves up to an impressive 3.03 acceptance rate.

## 5.4 WINDOW SIZE ABLATION

We investigate the effect of varying window size when training using the TOP objective. We pretrain 340M parameter models with the same setup as in Section 5.1, only changing the TOP window size with values 4, 16, 128, 1024, and 4096. We report the downstream benchmark results in Table 5, comparing each setup to the baseline NTP model as well. We observe varying results in each benchmark. We hypothesize that every task might benefit from different amounts of lookahead. All window sizes however outperform the NTP baseline.

Table 4: Average number of accepted tokens per forward pass when doing self-speculative decoding.

| Model | Size | Domain | | | |
|---|---|---|---|---|---|
| | | Wikipedia | Books | Code | Math |
| MTP | 340M | 2.145 | 1.883 | 2.232 | 2.208 |
| | 1.8B | 2.268 | 2.023 | 2.381 | 2.420 |
| | 7B | 2.392 | 2.090 | 2.460 | 2.488 |
| DS-MTP | 340M | 2.806 | 2.719 | 2.596 | 2.691 |
| | 1.8B | 2.963 | 2.821 | 2.754 | 2.805 |
| | 7B | 3.030 | 2.912 | 2.921 | 2.968 |
| TOP | 340M | 1.379 | 1.333 | 1.366 | 1.368 |
| | 1.8B | 1.472 | 1.414 | 1.485 | 1.507 |
| | 7B | 1.517 | 1.415 | 1.529 | 1.546 |

Table 5: Token order prediction window size ablation results on 340M models. We report the accuracy and perplexity on Lambada, the normalized accuracy on HellaSwag, ARC Challenge, PIQA, and SciQ, and the exact match score on NaturalQuestions Open and TriviaQA. Best scores are bolded.

| Window Size | Lambada | | HellaSwag | ARC | PIQA | SciQ | NQ Open | TriviaQA |
|---|---|---|---|---|---|---|---|---|
| | Acc. ↑ | PPL ↓ | N. Acc. ↑ | N. Acc. ↑ | N. Acc. ↑ | N. Acc. ↑ | E.M. ↑ | E.M. ↑ |
| 4 | 37.36 | 27.42 | 43.22 | 29.78 | 66.81 | 77.40 | **3.05** | **6.38** |
| 16 | **38.68** | **25.30** | 43.43 | **30.55** | 68.66 | 75.70 | 2.08 | 3.72 |
| 128 | 37.98 | 26.69 | **43.91** | 28.50 | **69.04** | 78.10 | 2.22 | 4.07 |
| 1024 | 36.95 | 27.80 | 43.74 | 30.12 | 67.85 | 76.60 | 2.66 | 4.15 |
| 4096 | 37.07 | 28.76 | 43.57 | 29.35 | 67.57 | **79.80** | 2.22 | 4.37 |
| NTP | 36.35 | 30.34 | 42.53 | 28.84 | 66.65 | 74.90 | 1.94 | 4.93 |

## 6 RELATED WORK

**Language Model Losses**   Many previous works have explored variations of the language modeling loss, most of them for use in training encoder models. Masked language modeling (MLM) randomly masks input tokens and trains the model to recover them from bidirectional context (Devlin et al., 2019). For example, T5 uses a denoising span-corruption objective in which contiguous spans are replaced by sentinel tokens and the model must reconstruct the spans (Raffel et al., 2020); this yields shorter target sequences and faster training. XLNet's permutation language modeling samples random autoregressive orderings to capture bidirectional dependencies while remaining autoregressive (Yang et al., 2019). Similarly, denoising autoencoder pretraining as in BART corrupts text (e.g., by shuffling sentences or infilling masked spans) and learns to reconstruct the original text (Lewis et al., 2020). UL2 (Tay et al., 2022) further unifies these ideas with a mixture-of-denoisers objective, interleaving various span- and prefix-corruption schemes to improve robustness across tasks. Retrieval-augmented models like RETRO add a nearest-neighbor retrieval step during pretraining, conditioning generation on retrieved document chunks to reduce perplexity and enable easy knowledge updates (Borgeaud et al., 2022). Other alternatives include replaced-token detection as in ELECTRA (Clark et al., 2020), where the model sees plausible substitutes in place of masked tokens and must identify which tokens were replaced.

**Multi-Token Prediction**   Next-token prediction (NTP), the standard training objective for language models, has been criticized for its limitations in long-range planning. The teacher-forcing approach may fail to learn an accurate next token predictor, which hinders the model's ability to plan beyond several tokens (Bachmann & Nagarajan, 2024). This issue motivates the exploration of alternative or auxiliary training objectives.

An enhancement to NTP is Multi-Token Prediction (MTP) (Gloeckle et al., 2024), an auxiliary objective designed to improve a model's look-ahead capabilities. MTP adds parallel heads that predict several future tokens simultaneously. This approach has shown significant improvements on planning-intensive tasks such as coding and has been adopted in state-of-the-art models like DeepSeek-V3 (DeepSeek-AI et al., 2024) and Ling-V2 (inclusionAI, 2025). As a complementary feature, MTP can also be used to speed up inference through self-speculative decoding.

The MTP framework has several variants. DeepSeek-V3 uses a sequential prediction mechanism with a small look-ahead window (N=2) to enhance decoding efficiency (DeepSeek-AI et al., 2024). Other approaches for multi-token awareness include converting NTP models to MTP models using register tokens (Gerontopoulos et al., 2025) and exploring parallel reasoning in a continuous space (Gozeten et al., 2025). Ahn et al. (2025) proposes to predict the joint probability of future tokens by carefully bottlenecking the architecture of the MTP heads.

## 7 LIMITATIONS

Due to limitations in compute and time, we are unable to pretrain on more data or larger models. The 7B models require 2 weeks of training time each on the 8xH200 node available to us. While our work demonstrates the potential of token order prediction for LLM pretraining, it remains to be seen whether TOP scales well to the standard of larger models and longer training runs of today.

## 8 CONCLUSION

In this paper, we propose Token Order Prediction (TOP) as a novel auxiliary training loss for LLM pretraining. Our approach addresses some limitations of Multi-Token Prediction (MTP) by replacing the difficult task of exact future token prediction with the more tractable objective of ranking upcoming tokens by their proximity. TOP requires only a single additional unembedding layer compared to MTP's multiple transformer layers, making it more parameter-efficient and scalable.

Based on the results of our general language modeling experiments across three model sizes (340M, 1.8B, and 7B parameters), TOP overall improves performance over NTP, MTP, and DS-MTP on standard NLP benchmarks. The method shows positive gains as parameter count grows, suggesting its potential value for larger-scale language models. Additionally, we further verify the power of the TOP objective by evaluating on the synthetic star graph pathfinding task. Here, TOP learns the correct look-ahead solution on graphs where NTP and MTP do not. Despite not being as effective as MTP for self-speculative decoding, these preliminary results indicate that TOP offers another promising direction for improving language model training through effective auxiliary objectives.

### REPRODUCIBILITY

To reproduce all of our experiments we provide the code repositories in the supplementary materials. The repositories include instructions on what commands to run and the configurations of each training run. We have detailed the needed datasets, training hyperparameters, and configurations for the experiments sections 5.1, 5.2, 5.3, and more specifically in Table 2. All datasets used are publicly accessible or readily able to be synthetically generated using the code. All trained models are also uploaded to public repositories for further research.

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
