# OpenReview forum: "Predicting the Order of Upcoming Tokens Improves Language Modeling"
_ICLR.cc/2026/Conference — Submitted to ICLR 2026_

### Official Review · Reviewer_UmzR · 2025-10-28

**Soundness:** 2
**Presentation:** 2
**Contribution:** 3
**Rating:** 2
**Confidence:** 3

**Summary:**

This paper introduces TOP, a new auxiliary training objective for language models that aims to enhance next-token prediction. TOP works by training models to rank upcoming tokens based on their proximity in the sequence. Unlike MTP, TOP uses only a single extra unembedding layer and relaxes the prediction task to ordering within a window. The authors pretrain transformer models of 340M, 1.8B, and 7B parameters on FineWeb-Edu using NTP, MTP, and TOP, evaluating them on eight standard NLP benchmarks and a synthetic star graph pathfinding task. Results show TOP outperforming NTP and MTP overall, especially at larger scales.

**Strengths:**

1. The idea of replacing exact multi-token prediction with a ranking-based objective is interesting and well-motivated. Compared to MTP, TOP's task relaxation to ordering makes it a more scalable auxiliary loss.
2. The TOP achieves consistent empirical gains, where experiments across eight diverse benchmarks and three model sizes are performed, and TOP improves over the NTP and MTP baselines.
3. TOP adds just one unembedding layer compared to MTP's multiple transformer heads, and scales better than MTP on non-coding tasks.

**Weaknesses:**

1. First, the presentation of this paper needs to be improved. In general, this paper introduces an intuitively effective and simple idea, i.e., using token order prediction to replace the much harder exact multiple token prediction. But the current draft makes it difficult to follow. For example, Fig 1 can be improved to illustrate how the proposed TOP works; Section 4 Method, instead of pure psudocode, the authors should also explain the working process of TOP; Figure 2, what do different color curves mean? I guess they indicate different positions, but it is better to be marked on the figure.
2. The empirical evaluation is narrow. The model is trained on limited pre-training budget, where 104B tokens are used for all sizes of models. For example, according to the scaling law, for 7B model, it is roughly 15 % of the compute used for comparable 7B models in the literature, and it is unclear whether TOP is still advantageous after full-scale training (e.g., 1T tokens).
3.  Except for the synthetic star graph, it would be good to provide other planning-intensive benchmarks results, which can better validate the task generalizability of TOP.

**Questions:**

1. Why was the window size set eqaul to the sequence length? How does downstream performance and training cost vary when changing the window size $W$?”
2. For the star graph task, MTP-4 succeeds on easier graphs but fails on harder ones. How about trying MTP with more heads?
3. What does the second row "NTP Loss" mean in Table 2? I think each column corresponds to NTP, MTP, and TOP loss respectively.

---

> ### Author Response · Authors · 2025-11-27
> **Author Rebuttal 1**
>
> Thank you very much for your thorough review of our paper. We have updated the attached PDf to the newest version of our paper. We attempt to address your concerns and answer your questions below:
>
> ## W1: The presentation of this paper needs to be improved
> We appreciate the constructive feedback on the paper's presentation. To address this, we have annotated the pseudocode in Section 4 with detailed comments to clarify the algorithmic steps. Additionally, we have updated Figure 2 with a legend as suggested. You are correct that the curves represent different lookahead positions. We included this to emphasize that the prediction task becomes progressively more difficult as the distance from the current token increases.
>
> ## W2: The empirical evaluation is narrow
> Thank you for the feedback. We acknowledge the value of further scaling; however, due to computational constraints, we are unable to train models larger than 104B tokens at this time. Our intention with the 7B model was to demonstrate that our method is scalable to the standard model size currently prevalent in academic research. Following the suggestion of Reviewer 1fMV, we will explicitly address this scaling limit in our Limitations section.
>
> ## W3: Other planning-intensive benchmarks results
>
> Thank you for the feedback. In addition to the synthetic star graph, we investigated empirical performance on downstream math and code tasks. We opted to fine-tune the models and evaluate them on HumanEval and GSM8K, specifically using a pipeline of continued pre-training on Python documents from Stack-Edu. However, we observed that the current fine-tuning configuration yields suboptimal results. We attribute this to the need for further hyperparameter optimization and code improvements in our fine-tuning recipe. For transparency, we provide the preliminary results on HumanEval below:
>
>
> | Size | Model | pass@2 (±)      | pass@8 (±)      | pass@16 (±)     | pass@32 (±)     | pass@64 (±)     |
> |------|-------|-----------------|-----------------|-----------------|-----------------|-----------------|
> | 1.8B | NTP   | 0.0136 ± 0.0064 | 0.0276 ± 0.0111 | 0.0354 ± 0.0128 | 0.0445 ± 0.0150 | 0.0549 ± 0.0178 |
> | 1.8B | MTP   | 0.0062 ± 0.0020 | 0.0220 ± 0.0069 | 0.0375 ± 0.0114 | 0.0556 ± 0.0165 | 0.0669 ± 0.0195 |
> | 1.8B | TOP   | 0.0138 ± 0.0055 | 0.0337 ± 0.0114 | 0.0465 ± 0.0140 | 0.0622 ± 0.0172 | 0.0793 ± 0.0212 |
> | 7B   | NTP   | 0.0069 ± 0.0025 | 0.0231 ± 0.0075 | 0.0380 ± 0.0112 | 0.0583 ± 0.0154 | 0.0869 ± 0.0217 |
> | 7B   | MTP   | 0.0219 ± 0.0075 | 0.0505 ± 0.0134 | 0.0714 ± 0.0167 | 0.0979 ± 0.0207 | 0.1280 ± 0.0258 |
> | 7B   | TOP   | 0.0223 ± 0.0072 | 0.0520 ± 0.0146 | 0.0678 ± 0.0173 | 0.0858 ± 0.0199 | 0.1098 ± 0.0241 |
>
> For math tasks, we performed continued pre-training on OpenMathInstruct-2 followed by fine-tuning on the GSM8K training set. While we observe improved performance compared to the code tasks, we believe the fine-tuning configuration remains suboptimal. Consequently, we consider these results too preliminary for formal inclusion in the paper without further optimization. However, for the sake of transparency, we present the current GSM8K results below:
>
> | Size | Model | Flexible EM (±) | Strict EM (±)   |
> |------|-------|-----------------|-----------------|
> | 1.8B | NTP   | 0.2540 ± 0.0120 | 0.2532 ± 0.0120 |
> | 1.8B | MTP   | 0.1888 ± 0.0108 | 0.2146 ± 0.0113 |
> | 1.8B | TOP   | 0.2623 ± 0.0121 | 0.2616 ± 0.0121 |

---

> > ### Author Response · Authors · 2025-11-27
> > **Author Rebuttal 2**
> >
> > ## Q1: Why was the window size set equal to the sequence length?
> >
> > We initially set the window size equal to the full sequence length to maximize the training signal available to the model. However, to investigate the impact of this hyperparameter, we conducted an ablation study using window sizes of 4, 16, 128, 1024, and 4096. The results are presented below:
> >
> > | Window Size | Lambada Acc. | Lambada PPL | HellaSwag N. Acc. | ARC N. Acc. | PIQA N. Acc. | SciQ N. Acc. | NQ Open E.M. | TriviaQA E.M. |
> > |-------------|--------------|-------------|--------------------|--------------|---------------|----------------|----------------|----------------|
> > | 4     | 37.36 | 27.42 | 43.22 | 29.78 | 66.81 | 77.40 | **3.05** | **6.38** |
> > | 16    | **38.68** | **25.30** | 43.43 | **30.55** | 68.66 | 75.70 | 2.08 | 3.72 |
> > | 128   | 37.98 | 26.69 | **43.91** | 28.50 | **69.04** | 78.10 | 2.22 | 4.07 |
> > | 1024  | 36.95 | 27.80 | 43.74 | 30.12 | 67.85 | 76.60 | 2.66 | 4.15 |
> > | 4096  | 37.07 | 28.76 | 43.57 | 29.35 | 67.57 | **79.80** | 2.22 | 4.37 |
> > | **NTP** | 36.35 | 30.34 | 42.53 | 28.84 | 66.65 | 74.90 | 1.94 | 4.93 |
> >
> > We observe varying performance across benchmarks, leading us to hypothesize that the optimal lookahead horizon is task-dependent. Crucially, however, all tested window sizes consistently outperform the NTP baseline.
> >
> > ## Q2: For the star graph task, how about trying MTP with more heads?
> > We have tested MTP-6 with six MTP heads, which would cover the maximum path length of 5. However, the model only achieved 19.35% accuracy on the test set with degree 5 path length 5 graphs, indicating that it still failed to learn the solution within the same training setup, despite having a higher parameter count.
> >
> > ## Q3: What does the second row "NTP Loss" mean in Table 2?
> > We appreciate the opportunity to clarify. We have updated Section 5.1 to explicitly define this metric. The NTP Loss row reports the loss associated with the standard next-token prediction task. For both MTP and TOP architectures, this corresponds to the loss calculated on the first prediction head. This metric allows us to evaluate how the auxiliary objectives affect the model's fundamental language modeling performance.

---

### Official Review · Reviewer_sV45 · 2025-10-28

**Soundness:** 3
**Presentation:** 3
**Contribution:** 3
**Rating:** 8
**Confidence:** 4

**Summary:**

This paper proposes Token Order Prediction (TOP), a novel auxiliary training objective for autoregressive language models that replaces the exact multi-token prediction (MTP) with a learning-to-rank formulation. TOP trains the model to rank all vocabulary tokens by their proximity to the current position within a configurable window. The authors implement TOP using a single additional linear layer and a ListNet-style ranking loss, making it far more parameter-efficient than MTP.

**Strengths:**

This paper is well structured and highly motivated. Replacing token identity prediction with ordinal proximity is a clever relaxation that retains lookahead signal while reducing optimization hardness.

TOP requires only one extra linear layer, in contrast to MTP's per-token transformer heads.

The evaluation is comprehensive, spanning multiple model sizes, diverse NLP benchmarks.

**Weaknesses:**

The observation that TOP achieves higher NTP training loss yet better downstream performance is interesting but underexplored. The authors hypothesize regularization but provide no ablation (e.g., varying TOP loss weight, early stopping comparisons) to confirm this.

The TOP target assigns scores to all vocabulary tokens, most of which do not appear in the window. This may create a highly sparse and noisy supervision signal.

**Questions:**

Please see Weaknesses.

---

### Official Review · Reviewer_LFeZ · 2025-10-31

**Soundness:** 2
**Presentation:** 2
**Contribution:** 2
**Rating:** 2
**Confidence:** 5

**Summary:**

This paper proposes Token Order Prediction (TOP) as a new auxiliary loss for large language model pretraining. Instead of predicting the exact identities of future tokens as in Multi-Token Prediction (MTP), TOP trains the model to rank future tokens by proximity using a listwise ranking loss (ListNet). The authors train models of 340M, 1.8B, and 7B parameters with NTP, MTP, and TOP objectives and evaluate them on eight NLP benchmarks as well as a synthetic star-graph pathfinding task. TOP reportedly outperforms both NTP and MTP while being simpler and more parameter-efficient.

**Strengths:**

1) Interesting reformulation of auxiliary objectives — The idea of relaxing MTP’s difficult prediction target into a ranking-based proximity task is conceptually appealing.

2) Simple and lightweight implementation — TOP only requires an additional unembedding layer and is compatible with existing transformer architectures.

3) Broad empirical evaluation — The experiments cover multiple model scales and both standard and synthetic benchmarks.

**Weaknesses:**

W1) **Figure 2 lacks clarity**

The figure is supposed to show that predicting farther tokens is harder, but there’s no legend or label for each position (t+1, t+2, …).
It’s unclear which curve corresponds to which distance, so the trend the authors claim isn’t visually evident.

A similar plot for the TOP objective would also help illustrate whether it really leads to smoother or easier training.

W2) **The claim that TOP is “easier” isn’t well supported**

The paper keeps describing TOP as an easier task than MTP but never shows evidence—no convergence plots, gradient analysis, or theoretical argument.

Without that, the claim feels more like intuition than proof.

W3) **Hyperparameter setup seems too shallow to trust**

Each model size only lists one fixed set of hyperparameters, with no sign of tuning or validation.
It’s unclear whether these settings are optimal or even comparable across NTP, MTP, and TOP.
This makes it hard to tell if the reported gains come from the new objective itself or just from a configuration that happens to work better for TOP.

W4) **Missing ablations and analysis**

There’s no variation of window size, loss weight, or ranking depth, and no look into how TOP affects learned representations.
Without these, it’s difficult to understand what the model is actually learning from the objective.

**Questions:**

Q1) **Combining MTP and TOP**
Have you tried using MTP and TOP together or at different training stages?
For example, applying TOP early and MTP later might be complementary.

Q2) **Speculative decoding and improvements**
TOP performs worse than MTP in self-speculative decoding (Table 4).
Is this a limitation of TOP or a mismatch with decoding efficiency?
Any ideas on improving it e.g., refining the ranking loss or combining TOP signals at inference?

---

> ### Author Response · Authors · 2025-11-27
>
> Thank you very much for your thorough review of our paper. We have updated the attached PDf to the newest version of our paper. We attempt to address your concerns and answer your questions below:
>
> ## W1: Figure 2 lacks clarity
> Thank you for the feedback. We have since added a legend to indicate the offsets of MTP heads from t+1,...,t+16. We have also taken your advice to add a comparison between MTP loss and TOP loss in the motivation section to give a clearer comparison.
>
> ## W2: The claim that TOP is “easier” isn’t well supported
> To support this claim empirically, we have updated Figure 2 to include the training loss trajectories. The figure compares the loss of the TOP head against the averaged loss across all MTP heads. As shown, the TOP head consistently achieves a lower loss value compared to the MTP heads throughout training, providing evidence that the objective is easier to optimize.
>
> ## W3: Hyperparameter setup seems too shallow to trust
> We understand the reviewer's concern. Due to limited computational resources, we were unable to perform an exhaustive hyperparameter search. Instead, we strictly adhered to the default configurations provided by the Flame framework (https://github.com/fla-org/flame). These configurations are based on well-established settings from prior work, such as DeltaNet (https://arxiv.org/pdf/2406.06484), which ensures that the reported gains are not due to arbitrary tuning.
>
> ## W4: Missing ablations and analysis
> We have included an ablation study on the window size in Section 5.4. We interpret the reviewer's mention of "ranking depth" as equivalent to the window size (the number of future tokens the model is tasked with ordering), and our analysis shows how this hyperparameter impacts performance. Regarding the loss weight, we maintained a fixed weight to isolate the impact of the ranking objective itself. The following are the results from the TOP window size ablation on the 340M model, also compared to baseline NTP:
>
> | Window Size | Lambada Acc. | Lambada PPL | HellaSwag N. Acc. | ARC N. Acc. | PIQA N. Acc. | SciQ N. Acc. | NQ Open E.M. | TriviaQA E.M. |
> |-------------|--------------|-------------|--------------------|--------------|---------------|----------------|----------------|----------------|
> | 4     | 37.36 | 27.42 | 43.22 | 29.78 | 66.81 | 77.40 | **3.05** | **6.38** |
> | 16    | **38.68** | **25.30** | 43.43 | **30.55** | 68.66 | 75.70 | 2.08 | 3.72 |
> | 128   | 37.98 | 26.69 | **43.91** | 28.50 | **69.04** | 78.10 | 2.22 | 4.07 |
> | 1024  | 36.95 | 27.80 | 43.74 | 30.12 | 67.85 | 76.60 | 2.66 | 4.15 |
> | 4096  | 37.07 | 28.76 | 43.57 | 29.35 | 67.57 | **79.80** | 2.22 | 4.37 |
> | **NTP** | 36.35 | 30.34 | 42.53 | 28.84 | 66.65 | 74.90 | 1.94 | 4.93 |
>
>
> ## Q1: Combining MTP and TOP Have you tried using MTP and TOP together or at different training stages?
> We did not explore this combination because MTP requires architectural changes, specifically moving some of the Transformer blocks into the MTP head. Even if we maintained the same number of Transformer blocks in the backbone and added extra blocks for the MTP head, it would result in an unfair comparison due to the discrepancy in total model size (parameter count) between the MTP model and other baselines.
>
> ## Q2: Speculative decoding and improvements TOP performs worse than MTP in self-speculative decoding
> We believe that TOP performs worse than MTP in self-speculative decoding because TOP cannot predict repetitive tokens; it can only predict the order of unique tokens. This is especially true for tasks that involve a lot of repetition, such as coding. To improve this, we need to modify the algorithm so that it handles the ordering of repetitive tokens as well.

---

### Official Review · Reviewer_1fMV · 2025-11-03

**Soundness:** 3
**Presentation:** 1
**Contribution:** 2
**Rating:** 2
**Confidence:** 4

**Summary:**

This paper proposes Token Order Prediction (TOP), an auxiliary training objective for LM that predicts the relative order and proximity of upcoming tokens rather than their exact values. The key insight is that Multi-Token Prediction (MTP), which attempts to predict exact future tokens, is too difficult. TOP instead uses a learning-to-rank loss where the target is a proximity-weighted bag-of-words vector: tokens appearing at distance d within a window W receive scores of W-d, with farther tokens scored as 0. This creates a soft probability distribution emphasizing near-future tokens. Architecturally, TOP adds only a single unembedding head (vs. MTP's multiple transformer layers). Experiments on 340M, 1.8B, and 7B parameter models show TOP consistently outperforms both NTP-only and NTP+MTP baselines across 8 standard NLP benchmarks, with gains increasing at larger scales. On a synthetic star graph pathfinding task designed to test look-ahead reasoning, TOP achieves 100% accuracy while MTP and NTP models fail on complex configurations. The paper also evaluates self-speculative decoding, where MTP performs better, but concludes TOP is a promising direction for improving LLM pretraining.

**Strengths:**

1. **Novel approach**: The paper proposes TOP as a theoretically sound middle ground between NTP (too myopic) and MTP (too difficult). The connection to learning-to-rank is elegant.

2. **Strong empirical results on specific tasks**: TOP consistently outperforms both baselines across most tasks, with particularly impressive results on the star graph task (100% accuracy where others fail) demonstrating genuine improvement in look-ahead reasoning capabilities.

**Weaknesses:**

1. **Missing critical baselines and ablations, while over-emphasizing obvious observations**:
   - Motivation section (page 3): Approximately two-thirds of a page is devoted to explaining Figure 2, which shows that MTP loss increases with prediction distance. This is an intuitive and expected result that does not warrant such extensive discussion.
   - MTP baseline incomplete: The paper only compares against Meta's MTP variant (multiple linear heads for MTP) but ignores DeepSeek's MTP architecture (sequential auxiliary transformer layers approach), which has become the dominant MTP design in recent LLM pretraining. This is a significant omission given DeepSeek-V3's prominence.
   - Window size ablation missing: The choice of window size W is a key hyperparameter for TOP, yet no ablation study investigates how performance varies with W. What is the sensitivity?

2. **Experimental setup raises questions about generalizability**:
   - Insufficient training data: Models are trained on only 100B tokens (a subset of FineWeb-Edu). For context, typical convergence for 7B models requires 1-2T tokens. Since TOP is an auxiliary loss, it's unclear whether the observed benefits persist in fully-converged models or only help in the under-trained regime. The paper should either use the full FineWeb-Edu dataset or explicitly discuss this limitation.
   - Task diversity limited: Table 2 focuses heavily on multiple-choice classification tasks. Given that the MTP usually emphasizes "look-ahead" reasoning, the paper should evaluate on code generation benchmarks (e.g., HumanEval, MBPP), where planning ahead is crucial. The star graph task is synthetic and may not reflect real-world look-ahead benefits.
   - Statistical rigor: Table 2 reports only single-run point estimates without error bars or confidence intervals. Many of the benchmarks (especially QA tasks) are known to have high variance. Including mean ± std across multiple seeds would strengthen the claims. Additionally, the absence of MMLU (a standard robust benchmark) is notable.

3. **Presentation quality issues**:
   - Figure 2: The color legend is missing, making it unclear which line corresponds to which future token prediction (t+1, t+2, etc.).
   - Mathematical notation: Vectors are not typeset in boldface (e.g., y, s should be **y**, **s**), making it difficult to distinguish vectors from scalars throughout the paper.
   - Algorithm 1: Critical symbols lack clear definitions. For example, `next[v]` (line 179) and the output `y` (line 178) are not explained in the caption or main text. The main text simply states "please refer to the pseudocode" without walking through the logic, forcing readers to reverse-engineer the algorithm.

**Questions:**

Regarding the model parameters comparison between TOP and MTP, a more detailed analysis should be presented. For example, the 7B model needs 4096*32000 extract parameters for TOP. while for MTP, it equivalent to how many additional token predictions?

---

> ### Author Response · Authors · 2025-11-27
> **Author Rebuttal 1**
>
> Thank you very much for your thorough review of our paper. We have uploaded a new version of our paper PDF. We attempt to address your concerns and answer your questions below:
>
> ## W1a: Motivation section
> Thank you for the feedback. We have cut this segment short, minimized the figure size and also added a better visualization of averaged MTP loss vs TOP loss to bring across the motivation better. We have maintained the commentary of the original MTP paper as to emphasize the weaknesses that we address with this paper.
>
> ## W1b: MTP baseline incomplete
> We did not manage to include it in time for initial review, however after the ICLR submission deadline, we have since completed the DeepSeek MTP baselines and added them to the paper. This includes 340M, 1.8B, 7B models and star graph experiments. Results are the following:
>
> | Size  | Model  | NTP Loss | Lambada Acc. | Lambada PPL | HellaSwag N. Acc. | ARC N. Acc. | PIQA N. Acc. | SciQ N. Acc. | Social IQa Acc. | NQ Open E.M. | TriviaQA E.M. |
> |-------|--------|----------|--------------|--------------|--------------------|--------------|---------------|----------------|-------------------|----------------|----------------|
> | 340M  | NTP    | 2.39 | 36.35 | 30.34 | 42.53 | 28.84 | 66.65 | 74.90 | 39.82 | 1.94 | 4.93 |
> | 340M  | MTP    | 2.47 | 35.32 | 35.31 | 42.73 | 29.86 | 66.49 | 77.40 | 39.00 | 2.35 | 2.55 |
> | 340M  | DS-MTP | 2.50 | 34.66 | 40.99 | 40.29 | 27.56 | 63.76 | 73.70 | 37.92 | 0.36 | 0.87 |
> | 340M  | TOP    | 2.40 | 37.07 | 28.76 | 43.57 | 29.35 | 67.57 | 79.80 | 39.00 | 2.22 | 4.37 |
> | 1.8B  | NTP    | 2.06 | 49.58 | 11.38 | 60.05 | 38.65 | 73.50 | 86.40 | 41.56 | 4.54 | 11.85 |
> | 1.8B  | MTP    | 2.14 | 47.93 | 13.69 | 58.29 | 40.61 | 73.07 | 87.20 | 42.12 | 4.46 | 15.98 |
> | 1.8B  | DS-MTP | 2.14 | 48.71 | 13.32 | 57.48 | 40.44 | 71.87 | 86.40 | 42.84 | 4.21 | 12.06 |
> | 1.8B  | TOP    | 2.08 | 50.34 | 11.19 | 60.45 | 42.32 | 74.16 | 87.90 | 42.53 | 5.37 | 18.93 |
> | 7B    | NTP    | 1.88 | 55.89 | 7.97 | 67.44 | 45.65 | 76.99 | 88.60 | 44.37 | 7.31 | 24.28 |
> | 7B    | MTP    | 1.95 | 53.13 | 8.99 | 65.85 | 45.56 | 75.73 | 89.30 | 44.11 | 7.40 | 23.36 |
> | 7B    | DS-MTP | 1.96 | 55.62 | 8.52 | 66.03 | 44.37 | 75.79 | 88.70 | 43.76 | 6.57 | 18.54 |
> | 7B    | TOP    | 1.89 | 57.03 | 7.64 | 68.73 | 46.42 | 76.39 | 91.60 | 43.91 | 7.70 | 30.90 |
>
> From these results we see that TOP remains a competent advancement from both MTP and DeepSeek MTP training objectives.
>
> ## W1c: Window size ablations
> Thank you for the suggestion. We have run the ablations for window size of 4, 16, 128, 1024, and 4096. Results are the following:
>
> | Window Size | Lambada Acc. | Lambada PPL | HellaSwag N. Acc. | ARC N. Acc. | PIQA N. Acc. | SciQ N. Acc. | NQ Open E.M. | TriviaQA E.M. |
> |-------------|--------------|-------------|--------------------|--------------|---------------|----------------|----------------|----------------|
> | 4     | 37.36 | 27.42 | 43.22 | 29.78 | 66.81 | 77.40 | **3.05** | **6.38** |
> | 16    | **38.68** | **25.30** | 43.43 | **30.55** | 68.66 | 75.70 | 2.08 | 3.72 |
> | 128   | 37.98 | 26.69 | **43.91** | 28.50 | **69.04** | 78.10 | 2.22 | 4.07 |
> | 1024  | 36.95 | 27.80 | 43.74 | 30.12 | 67.85 | 76.60 | 2.66 | 4.15 |
> | 4096  | 37.07 | 28.76 | 43.57 | 29.35 | 67.57 | **79.80** | 2.22 | 4.37 |
> | **NTP** | 36.35 | 30.34 | 42.53 | 28.84 | 66.65 | 74.90 | 1.94 | 4.93 |
>
> We observe varying results in each benchmark. We hypothesize that every task might benefit from different amounts of lookahead. All window sizes however outperform the NTP baseline.
>
> ## W2a: Insufficient training data
> We understand your concern and agree that our setup does not equate to full large training runs of modern LLMs. However, we do not have the compute nor time to train on 1T tokens or more. Our 7B training runs with 104B tokens took 2 weeks each with the best compute setup we have at hand. As per your suggestion, we will add a limitation section to address this.

---

> > ### Author Response · Authors · 2025-11-27
> > **Author Rebuttal 2**
> >
> > ## W2b: Task diversity limited
> > Thank you for the suggestion. To address this, we have opted to finetune models for code and math, and evaluate on HumanEval and GSM8K. Our current pipeline for code is that we continue pretraining on python documents on stack-edu. However, we have decided not to include the results as they are very weak. We suspect our finetuning procedure to be insufficient and suboptimal, and we will have to improve it before including results in this task. Here are the results on HumanEval:
> > | Size | Model | pass@2 (±)      | pass@8 (±)      | pass@16 (±)     | pass@32 (±)     | pass@64 (±)     |
> > |------|-------|-----------------|-----------------|-----------------|-----------------|-----------------|
> > | 1.8B | NTP   | 0.0136 ± 0.0064 | 0.0276 ± 0.0111 | 0.0354 ± 0.0128 | 0.0445 ± 0.0150 | 0.0549 ± 0.0178 |
> > | 1.8B | MTP   | 0.0062 ± 0.0020 | 0.0220 ± 0.0069 | 0.0375 ± 0.0114 | 0.0556 ± 0.0165 | 0.0669 ± 0.0195 |
> > | 1.8B | TOP   | 0.0138 ± 0.0055 | 0.0337 ± 0.0114 | 0.0465 ± 0.0140 | 0.0622 ± 0.0172 | 0.0793 ± 0.0212 |
> > | 7B   | NTP   | 0.0069 ± 0.0025 | 0.0231 ± 0.0075 | 0.0380 ± 0.0112 | 0.0583 ± 0.0154 | 0.0869 ± 0.0217 |
> > | 7B   | MTP   | 0.0219 ± 0.0075 | 0.0505 ± 0.0134 | 0.0714 ± 0.0167 | 0.0979 ± 0.0207 | 0.1280 ± 0.0258 |
> > | 7B   | TOP   | 0.0223 ± 0.0072 | 0.0520 ± 0.0146 | 0.0678 ± 0.0173 | 0.0858 ± 0.0199 | 0.1098 ± 0.0241 |
> >
> > For math, we continued pretraining on OpenMathInstruct-2 and then finetuned on the train set of GSM8K. Here, we see better results, but we still think the finetuning code is flawed and that we cannot include these tasks before fixing it. Here are the results on GSM8K nonetheless:
> > | Size | Model | Flexible EM (±) | Strict EM (±)   |
> > |------|-------|-----------------|-----------------|
> > | 1.8B | NTP   | 0.2540 ± 0.0120 | 0.2532 ± 0.0120 |
> > | 1.8B | MTP   | 0.1888 ± 0.0108 | 0.2146 ± 0.0113 |
> > | 1.8B | TOP   | 0.2623 ± 0.0121 | 0.2616 ± 0.0121 |
> >
> >
> > ## W2c: Statistical rigor
> > We understand your concern. However, we also again do not have enough compute nor time to execute runs with multiple seeds. As for MMLU, we decided not to include it in the paper as our models are not trained on enough data to answer the highly field-specific questions inside MMLU, which results in near random (~25%, MMLU contains 4-choice questions) results:
> > | Size | Model | Accuracy |
> > |------------|-----------------|----------|
> > | 340M | NTP | 24.63 |
> > | 340M | MTP | 23.21 |
> > | 340M | TOP | 23.67 |
> > | 1.8B | NTP | 24.46 |
> > | 1.8B | MTP | 23.34 |
> > | 1.8B | TOP | 25.71 |
> > | 7B   | NTP | 25.25 |
> > | 7B   | MTP | 25.27 |
> > | 7B   | TOP | 26.15 |
> >
> > ## W3: Presentation quality issues
> > Thank you very much for your feedback on our presentation. We have added a legend to the figure in Section 3 Motivation to indicate the MTP losses of each head. We have revised all the mathematical notation with different typesets for scalars, vectors, and also functions. We have written clearer comments in Algorithm 1. Please let us know if there is still anything we can still improve.
> >
> > ## Q: Parameter gain for TOP and MTP
> > We have added this analysis in Section 4:
> > > Specifically, while a TOP head requires $DV$ extra parameters, $N$ number of MTP heads require $N(4D^2 + 12D^2 + 2D)$ assuming a standard transformer block with MLP hidden size $4D$ and 2 RMSNorms.

---

### Meta-Review · Area_Chair_Jdsm · 2026-01-04

**Summary:**

This paper presents a auxiliary loss, Token Order Prediction (TOP), that is a relaxed objective compared with the existing Multi-Token Prediction (MTP) loss. The paper presents pretraining results on models of multiple scales to validate the empirical performance gains.

The reviewers raised concerns on presentation clarity, missing ablations/baselines, limited training scale (e.g., w.r.t. number of tokens) and narrow evaluations. During the rebuttal, the authors have provided substantial additional experimental results to address many of the concerns.

I acknowledge that the simplicity of the proposed method is a practical advantage. However, given the shallow technical depth, it would make the paper substantially stronger with more convincing empirical evidence or additional theoretical understanding.

**Reviewer Concerns:**

# Reviewer 1fMV

The rebuttal provided additional results regarding the initial concerns on missing baselines (DS-MTP) and ablation on window size. However, there lacks a way to determine the window size. This leaves the question whether the comparison with the baseline is fair due to the additional training budget for multiple window sizes. The authors have also revised the paper to improve the presentation quality, and provided technical details on the parameter overhead compared with MTP, which should address the initial concerns.

The authors clarified the single-seed experiments (statistical rigor) due to limited computation resources, which is reasonable. The authors acknowledged that results on coding and math may not be reliable. I agree with the reviewer that such results may be valuable in understanding the model's true capability.

# Reviewer LFeZ

The authors have revised the paper to address the initial concerns on presentation clarity. The authors clarifies the hyperparameter selection criteria, which sounds reasonable. However, addressing this issue would indeed make the paper more convincing. The authors provided ablation on window size, where the evidence does not reveal a guideline to select the window size -- this could cause substantial overhead in tuning, and thus is a remaining concern.

The authors also acknowledged the limitation of the proposed TOP in speculative decoding.

# Reviewer sV45

No rebuttal has been provides.

# Reviewer UmzR

The reviewer mentioned limited token budget for the 7B model, which is a valid concern. The authors acknowledged the limitation due to compute resources.

Planning-intensive benchmarks: the authors instead provided results on coding and math. See comments above for the remaining concern.

Sequence length and window size: the authors provided ablation on various window sizes. See comments above on the remaining concern.

**Reviewer Scores:**

The reviewers giving a "reject" initial score may moderately increase their score, due to the improved presentation and the new results with additional baselines. However, the overall final score may still fall short of the acceptance threshold due to remaining issues on coding/math/planning benchmarks, limited empirical evidence & ambiguity in hyperparameter selection.

---

### Decision · Program_Chairs · 2026-01-26

Reject